# Numerical Simulation of Static Seal Contact Mechanics Including Hydrostatic Load at the Contacting Interface

De Huang [1,*], Xiang Yan [2], Roland Larsson [1] and Andreas Almqvist [1]

1   Division of Machine Elements, Luleå University of Technology, 97187 Luleå, Sweden;
    roland.larsson@ltu.se (R.L.); Andreas.Almqvist@ltu.se (A.A.)
2   Saint-Gobain, Bristol, RI 02809, USA; Xiang.Yan@saint-gobain.com
*   Correspondence: de.huang@ltu.se

**Abstract:** A finite element model of a static seal assembled in its housing has been built and is utilized to study how the seal deforms under varying loading conditions. The total contact load on the sealing surface is balanced by the sealed fluid pressure and the friction between the seal and the housing sidewall perpendicular to the sealing surface. The effect of the sealed fluid pressure between the sealing surfaces was investigated and the simulation showed that the surface profile is distorted due to the hydrostatic pressure. We study the distorted contact profile with varying sealed fluid pressure and propose five parameters to describe the corresponding contact pressure profile. One of these parameters, overshoot pressure, a measure of the difference between maximum contact pressure and the sealed fluid pressure, is an indicator of sealing performance. The simulations performed show different behaviors of the overshoot pressure with sealed fluid pressure for cosinusoidal and parabolic surfaces with the same peak to valley (PV) value.

**Keywords:** static seal; contact mechanics; hydrostatic load

## 1. Introduction

A static seal is an important machine element. By definition, a static seal remains stationary and is subjected to no movement under operation. The installation of a static seal requires a pre-tension to creates a tight fit on the sealing surface between the seal and shaft [1]. During operation, the sealed fluid fills the space at the high-pressure side of the seal, and if no leakage occurs, the fluid is in a hydrostatic state and the pressure equals the pump pressure everywhere. Considering the ease of seal installation, the pre-tension is limited and it is typically much smaller than the required load for fluid sealing. To provide sufficient contact force at the sealing surface, it is common practice to utilize the sealed fluid pressure itself as a source of contact force to ensure the tightness of the seal.

Seals lose their function when the contact between the sealing surfaces is lost. Misalignment of the mating parts, material extrusion, elastic leak, thermal expansion/shrinkage, and wear damage are examples of common failure modes ([2–4]). The research interest in this study focuses on the sealing performance of the static seal under standard temperature and pressure (STP) condition (Since 1982, STP is defined as a temperature of 273.15 K (0 °C, 320 °F) and an absolute pressure of $10^5$ Pa (100 kPa, 1 bar)). Seal leakage can significantly increases when the sealed fluid pressure surpasses the contact pressure without material failure [4]. Results have shown, see Reference [5], that the planar sealing surface with long contact length is prone to both extrusion damage and leakage. Therefore, anti-extrusion seal design utilizes non-planar geometry both before and after the contacting part of the surface [6]. A non-planar geometry of the contacting surface has multiple benefits besides decreasing extrusion. For instance, by concentrating the total load on a smaller area, the higher contact pressure is achieved under the same total load condition. However, the non-planar surface profile allows pressured fluid to penetrate between the sealing surfaces

and the contact state changes with the sealed fluid pressure, therefore, increase the risk of leakage.

The seal assembly and the loading condition are closely related to the performance of the seal, and they determine the contact area and leakage paths. The contact between deformable solids has been extensively studied and, for dry conditions, there are analytical solutions available. See for example References [7–9], for analytical solutions for regular sinusoidal surface patterns in 1D and 2D. Numerical solutions also exist for random-surface patterns based on the statistical properties of the surface roughness, see References [10–13]. When the sealed fluid pressure interacts with the housing, as a source of contact load, it can affect the global contact length [14]. In addition, the increased sealed fluid pressure can propagate the fluid front within the contact interface, as observed in both the experiment and finite element simulation [15]. This phenomenon can be explained by the linear elastic fracture mechanics (LEFM) theory [16]. So the fluid load must be considered for the cases where the fluid is present between the sealing surfaces [17,18]. In Reference [18], it was shown that a fully coupled approach considering the two-way fluid-structure interaction, between the sealing surfaces, is preferred compared with a sequential solution procedure, in which the equations are solved independently and the solution is obtained through iteration until convergence is reached.

Unlike the rubber seal, which is easy to deform and allows an installation with large overfitting, the non-elastomer seal made out of a much stiffer polymer or metal is difficult to compress. A typical static non-elastomer seal assembly is shown in Figure 1. In the current configuration a gap exists, between the seal ring and the housing, so that the sealed fluid pressure at all times contributes to the contact force. It is of particular interest to know the effect of the sealed fluid pressure on leakage performance. However, according to the authors' knowledge, very few studies have been attempted to address this practical sealing challenge.

In this paper, we study the contact state for a widely used seal assembly, and its configuration is depicted in Figure 1. In this configuration, the total contact load increases with the sealed fluid pressure. Meanwhile, the fluid can fill the space at the high-pressure side and provide a hydrostatic lift on the bottom surface of the seal. To this end, a new, finite-element based, contact-mechanics framework was developed. The model includes the fluid load acting on the seal ring at the high-pressure side wall, the top wall and inside the interface between the contacting surfaces. Numerical simulations for varying sealed fluid pressure and friction conditions are conducted, and five parameters that describe the contact profile are presented and discussed.

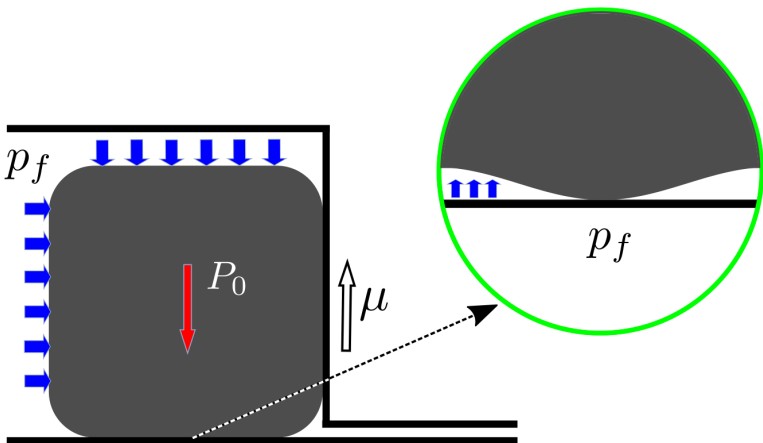

**Figure 1.** Cross-section schematics of a static seal in its housing.

## 2. Problem Set-Up

Figure 1, shows the cross-section schematics of a static seal in its housing. The seal has a square-shaped cross-section with a side length of $\lambda$ and a non-planar bottom. The origin

of the coordinate system for the finite element analysis is defined at the midpoint of the bottom surface and the $y$-axis is parallel with the housing sidewall, see Figure 2. The pretension with a line load density of $P_0$ in the $y$-direction provides the initial sealing force against the shaft. The sealed fluid acts on the top and the left side as well as on the bottom side, via the interface between the sealing surfaces. The bottom surface can be separated into three regions based on their contacting states. At the high-pressure side, before solid-solid contact happens, the interface is filled with the sealed fluid. Following the fluid penetrated region $\partial \Omega_b$, solid-solid contact happens, and this region is defined as $\partial \Omega_c$. After the contact region, at the low-pressure side, the seal and the shaft are separated. The roughness on the contacting interface is ignored in the current analysis, because its length scale is much smaller than the non-planar curvature.

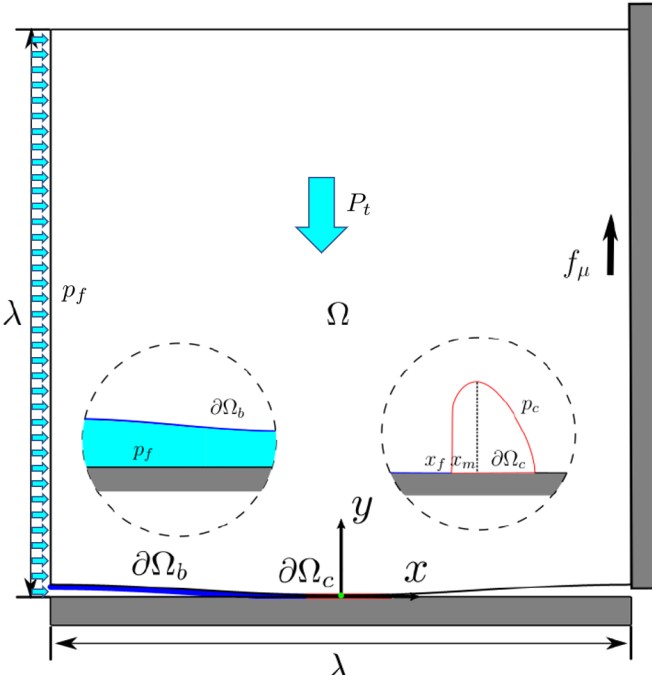

**Figure 2.** Details of the problem set-up and FE -model, where $\partial \Omega_b$ is the part of the boundary, of the bottom surface, on which the sealed fluid pressure $p_f$ acts, $\partial \Omega_c$ is the contact zone where the contact pressure $p_c$ acts, $x_f$ is the location of fluid front, that is, the boundary between the sealed fluid and the solid contact zone, and $x_m$ is the location where the maximum contact pressure occurs. The friction coefficient between the seal and the vertical sidewall is $\mu$. The total load at vertical direction is $P_t$ and sealed fluid pressure $p_f$ is acting at the left vertical side of the seal.

For the sake of simplicity, we consider the problem under plane strain conditions. Two different surface profiles of the non-planar bottom surface, cosinusoidal and parabolic, which exhibit the same peak to valley value (PV) are of interest for this study. More precisely, these are cosinusoidal

$$y(x, z) = \Delta \cos(2\pi x / \lambda), \tag{1}$$

and parabolic

$$y(x, z) = 8\Delta x^2 / \lambda^2, \tag{2}$$

profiles, where $\Delta$ is the amplitude of the non-planar form. The contacting surface between the vertical wall of the seal and the housing is assumed to be leak-tight, and is modeled with a static friction coefficient $\mu$. Bottom contact surface is assumed to be frictionless. A linear elastic material is assigned to the seal, and the shaft and housing which have

higher elastic modulus than the seal, is modeled as rigid bodies. Under the plane strain condition, we have

$$\epsilon_z = 0, \quad \sigma_z = \nu(\sigma_x + \sigma_y), \tag{3}$$

and Hooke's law can be formulated as

$$\epsilon_x = \frac{1}{E}\left(\left(1-\nu^2\right)\sigma_x - \nu(1+\nu)\sigma_y\right), \tag{4}$$

$$\epsilon_y = \frac{1}{E}\left(\left(1-\nu^2\right)\sigma_y - \nu(1+\nu)\sigma_x\right), \tag{5}$$

$$\gamma_{xy} = \frac{2(1+\nu)}{E}\tau_{xy}, \tag{6}$$

where $E$ is the elastic modulus of the seal and $\nu$ is Poisson's ratio. The system to be solved is described by the following equations:

$$\frac{\partial\sigma_x}{\partial x} + \frac{\partial\tau_{xy}}{\partial y} = 0 \quad \text{in} \quad \Omega, \tag{7}$$

$$\frac{\partial\sigma_y}{\partial y} + \frac{\partial\tau_{xy}}{\partial x} = 0 \quad \text{in} \quad \Omega, \tag{8}$$

$$\frac{\partial^2\epsilon_x}{\partial y^2} + \frac{\partial^2\epsilon_y}{\partial x^2} = \frac{\partial^2\gamma_{xy}}{\partial x\partial y} \quad \text{in} \quad \Omega, \tag{9}$$

$$\int_{-\frac{\lambda}{2}}^{\frac{\lambda}{2}} \sigma_y|_{y=\lambda}dx = -(P_0 + p_f\lambda), \tag{10}$$

$$\sigma_x|_{x=-\frac{\lambda}{2}} = -p_f, \tag{11}$$

$$\int_{-\frac{\lambda}{2}}^{\frac{\lambda}{2}} \tau_{xy}|_{x=\frac{\lambda}{2}}dy = -\mu\int_{-\frac{\lambda}{2}}^{\frac{\lambda}{2}} \sigma_x|_{x=\frac{\lambda}{2}}dy, \tag{12}$$

$$u_x|_{x=\frac{\lambda}{2}} = 0, \tag{13}$$

$$(\boldsymbol{\sigma}\cdot\boldsymbol{n})|_{x\leq x_f} = -p_f \quad \text{on} \quad \partial\Omega_b, \tag{14}$$

$$|\boldsymbol{\sigma}\cdot\boldsymbol{n}| > p_f \quad \text{on} \quad \partial\Omega_c \quad \text{if} \quad x < x_m. \tag{15}$$

Equations (10)–(15) summarize the boundary conditions. The total applied load along the $y$-direction is given by Equation (10), and the load balance between sealed fluid pressure, on the left vertical sidewall and the horizontal normal stress on the seal ring, is given by Equation (11). Equations (12) and (13) define the boundary conditions for the stresses and displacements for the vertical interface between the seal and the housing. The seal displacement in the $x$-direction is $u_x$. Equation (14) defines the hydrostatic loading condition on the boundary $\partial\Omega_b$, which is separated from $\partial\Omega_c$ at $x_f$, which is the location of the fluid front. Equation (15) defines the contact load constraint such that the solid-solid contact pressure for all the points $x < x_m$ are required to be higher than the sealed fluid pressure $p_f$, with $x_m$ as the position where the contact pressure reaches its maximum value. The difficulty in solving the above system lays in that the location of the fluid front, $x_f$ is not known beforehand. In the present work, the system of equations, comprised by Equations (3)–(15), is set up and solved in COMSOL 5.4, and the penalty method is utilized for the contact constraint with a contact tolerance of $10^{-6}\lambda$.

The solution domain is initially discretized using triangular quadratic-order elements, as shown in Figure 3. Numerical solutions for three sets of meshes, with decreasing maximum element size, are computed for the case with $p_f = 0$ and $\mu = 0$. The bottom contacting surface is meshed with maximum element sizes equal to $2 \times 10^{-3}\lambda$, $1 \times 10^{-3}\lambda$ and $5 \times 10^{-4}\lambda$, with an element growth rate of 1.05. This resulted in meshes with 8367,

16,467 and 32,253 regular- and 682, 1210 and 2236 edge elements, respectively. Westergaard gave an analytical solution for a periodic cosinusoidal surface contact with a rigid flat surface, for linear elastic material under the half-space assumption . The minimum pressure required to flatten the periodic consinousoidal surface is [7]:

$$p_s = \frac{\pi E}{1 - \nu^2} \left( \frac{\Delta}{\lambda} \right), \tag{16}$$

where $E$ is the elastic modulus for the seal and $\nu$ is the Poisson's ratio. The Westergaard solution can be reproduced by the current model with $\mu = 0, p_f = 0$, if the boundary condition in (13) is replaced by the periodic condition

$$u_x\big|_{\left(x=\frac{\lambda}{2}\right)} = u_x\big|_{\left(x=-\frac{\lambda}{2}\right)}. \tag{17}$$

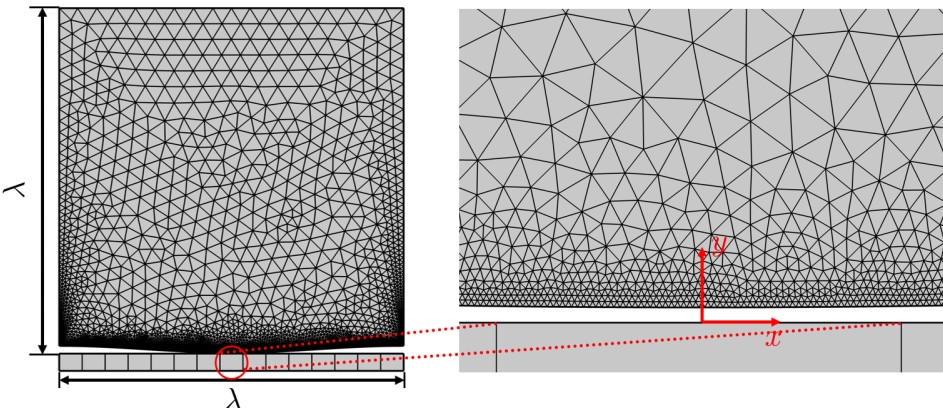

**Figure 3.** Finite element mesh with 16,467 regular elements and 1210 edge elements at the bottom surface.

As a benchmark, the contact length and the maximum contact pressure using these settings, for 3 different meshes, are compared with the corresponding Westergaard solution. The solution is considered converged when the residual is smaller than $10^{-6}$. For the mesh with 16,467 regular elements and 1210 edge elements, the maximum relative error is less than 1% for the maximum contact pressure and 2.5% for the half contact length. To validate the effect of the sealed fluid pressure, the numerical solution, for the cosinusoidal bottom surface profile given by Equation (1) and $2\Delta/\lambda = 1/50$, with uniform fluid pressure penetrated from both sides of the contacting surfaces is compared with the analytical solution presented in Reference [19]. In this case, the numerical solution is acquired using the boundary condition (17), and changing the boundary condition (15) to

$$|\boldsymbol{\sigma} \cdot \boldsymbol{n}| > p_f \quad \text{on} \quad \partial\Omega_c. \tag{18}$$

The contact mechanics problem with load control boundary condition is non-linear, therefore, the convergence of the solution procedure is highly dependent on the initial guess. A good strategy to increase the stability of the solution procedure is to increase the force and pressure load from zero to the target value gradually. To this end, a ramping-up function is added to both the boundary conditions and the loads

$$f_{ramp} = \frac{1}{2}(1 - \cos(\pi\theta)), \quad \theta \in [0, 1], \tag{19}$$

and the boundary conditions (10), (11), (14) and (15) become

$$\int_{-\frac{\lambda}{2}}^{\frac{\lambda}{2}} \sigma_y\big|_{y=\lambda} dx = -(P_0 + p_f\lambda)f_{ramp}(\theta), \tag{20}$$

$$\sigma_x|_{x=-\frac{\lambda}{2}} = -p_f f_{ramp}(\theta), \tag{21}$$

$$(\boldsymbol{\sigma} \cdot \boldsymbol{n})|_{x \le x_f} = -p_f f_{ramp}(\theta) \quad \text{on} \quad \partial \Omega_b, \tag{22}$$

$$|\boldsymbol{\sigma} \cdot \boldsymbol{n}| > p_f f_{ramp}(\theta) \quad \text{on} \quad \partial \Omega_c \quad \text{if} \quad x < x_m. \tag{23}$$

Meanwhile, a spring foundation is applied on the top surface, with a total spring constant $k_{tot} = k_0(1 - f_{ramp}(\theta))2^{-10f_{ramp}(\theta)}$, to provide extra damping on the loading boundary condition. The spring parameter $k_0$, when normalized with the material elastic modulus, is chosen as 0.01 to optimize the convergence speed. The simulation is performed by starting from $\theta = 0$, which gives 0 boundary load, 0 fluid load, and maximum spring constant, and it is then gradually increased to 1, so that the loadings and constraints equal to the target.

## 3. Results

Figure 4 depicts the normal stress and the gap height between the contacting surfaces for the cases with the total line load density in the vertical direction equals

$$P_t(p_f) = P_0 + p_f \lambda, \tag{24}$$

for the cosinusoidal profile with $2\Delta/\lambda = 1/50$. The sidewall friction is ignored, that is, $\mu = 0$, and the Poisson ratio of the material is $\nu = 0.33$. The results are depicted in non-dimensional form. The scaling factor for the contact length is $\lambda$ and the gap height between the contacting surfaces is scaled with $\Delta$. The displacement and strain for the current problem are independent of the elastic modulus [20]. By Hooke's law, the stress is linearly dependent of $E$, meaning that scaling the stress with $E$ makes it independent of the elastic modulus. To this end, the stress is normalized with $p_s$, defined in Equation (16).

The solid lines in Figure 4 represent the stress and the gap height between the contacting surfaces for the cases with the sealed fluid pressure on $\partial \Omega_b$. On the other hand, the dashed lines in Figure 4 represent the cases when there is no fluid between the contacting surfaces and the total line load density along $y$-direction equals $P_t$. The difference between the solid and dashed profiles with the same $P_t$ value reveals multiple effects of the hydrostatic pressure between the contacting surfaces, some of which are discussed in the following. For instance, the presence of the sealed fluid pressure tends to push the seal towards the low-pressure side. This causes the reduction of maximum contact pressure and contact length compared with the dry contact cases when $P_t$ values are identical, as seen in Figure 4a,b. The points defining the boundaries of the contact are both moving away from their initial positions ($P_t = P_0$), where the initial pre-tension $P_0$, for the results presented in Figure 5, is chosen such that the coordinate of the first solid-solid contact point at the high-pressure side $x_0$ is $-0.054\lambda$ when $p_f = 0$. The total contact length is initially increasing when the sealed fluid pressure is increased. In the gap, between the seal and the shaft at the high-pressure side in front of the fluid front $x_f$, the normal stress equals the hydrostatic pressure ($\sigma_y/p_s = p_f/p_s$). The interfacial fluid at the high-pressure side provides load support in the vertical direction and suppresses the seal deformation at the high-pressure side. Meanwhile, there is no fluid support between the contacting surfaces at the low-pressure side, the deformation here is close to the dry contact cases, as seen in the dashed lines in Figure 4b. Therefore, the change of the fluid front location, $x_f$, is not as large as the change of the last contact point at the low-pressure side. The location of the maximum contact pressure shifts from the origin of the defined coordinate system and moves towards the low-pressure side, because of the fluid load acting from the high-pressure side. For the dry contact cases, the horizontal reaction force from the vertical sidewall at $x = \lambda/2$ forces the contacting surface to tilt towards the high-pressure side, which results in smaller gap height there.

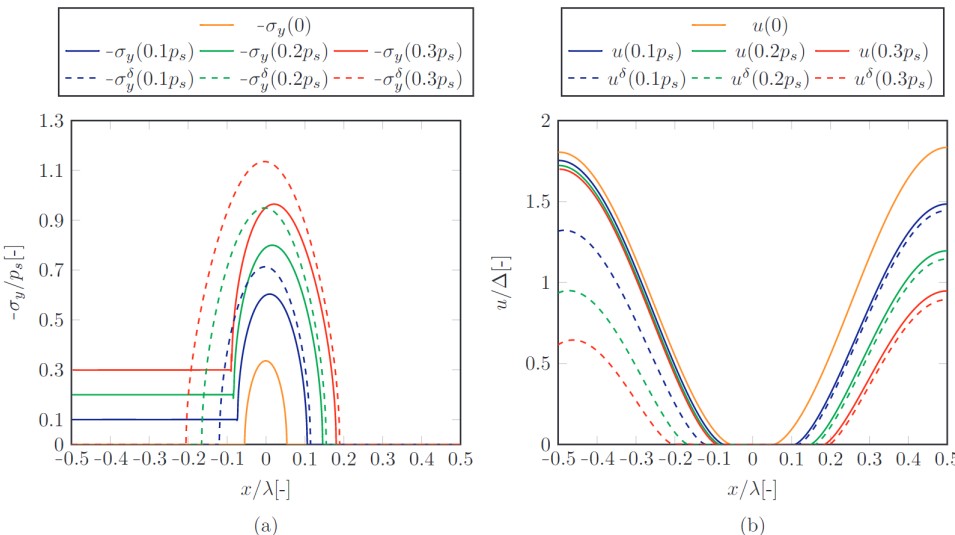

(a)　　　　　　　　　　　　　　　　　　　　(b)

**Figure 4.** The stress of the contacting surface and gap height between the contacting surfaces. (**a**) Normalized $y$-direction stress at the contacting surfaces as a function of the sealed fluid pressure $p_f$. The continuous lines represent $y$-direction stress $\sigma_y = \sigma_y(p_f)$ for the situation when there is fluid inside the interface between sealing surfaces, and the total line load density in the vertical direction is $P_t(p_f) = P_0 + p_f\lambda$. The dashed lines represent the $y$-direction stress for the dry contact case under the total line load density $P_t(p_f)$, for which the $y$-direction stress is $\sigma_y^\delta = \sigma_y^\delta(p_f)$. (**b**) The gap height between the contacting surfaces. The continuous lines represent the gap height between the contacting surfaces $u = u(p_f)$ for the situation when there is fluid between the sealing surfaces. The dashed lines represent the gap height between the contacting surfaces for the dry contact case under the total line load density $P_t(p_f) = P_0 + p_f\lambda$ along $y$-direction, for which the gap height between the contacting surfaces is $u^\delta = u^\delta(p_f)$.

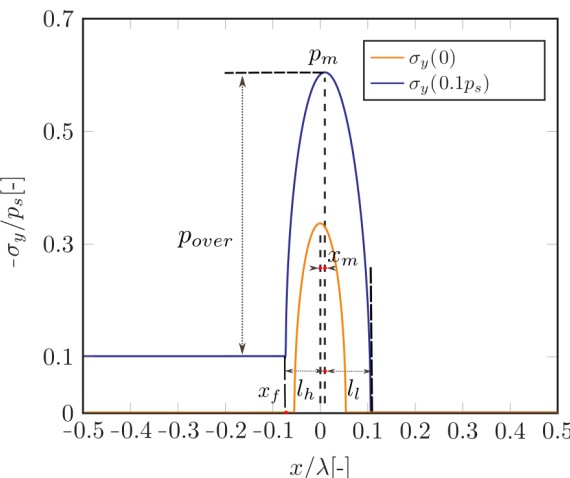

**Figure 5.** An overview and definition of the five performance parameters used in the analysis.

Five performance parameters, as shown in Figure 5, are defined to describe the contact profile of the current one-sided fluid penetration case.

- $p_m$: the maximum contact pressure.

- $x_m$: the maximum contact pressure location.

- $l_h$: length of contact at the high-pressure side, that is, from the location of the fluid front $x_f$ to $x_m$.

- $l_l$: length of contact at the low-pressure side, that is, from $x_m$ to the last contact point.

- $p_{over}$: overshoot pressure, that is, the difference between the maximum contact pressure and the sealed fluid pressure; $p_{over} = p_m - p_f$. The overshoot pressure $p_{over}$ equals to the maximum contact pressure, $p_m$, when the sealed fluid pressure, $p_f$, is 0. As $p_f$ increases, $p_{over}$ decreases and when $p_{over}$ reaches to 0, leakage occurs.

The variation of the performance parameters $p_m$, $x_m$, $l_h$ and $l_l$ with $p_f$ can be found in Figure 6. The maximum contact pressure increases with the sealed fluid pressure, as shown in Figure 6a. A fraction of the total load is supported by the friction at the vertical sidewall, and with the increased sidewall friction coefficient, the maximum contact pressure decreases. The location of the maximum contact pressure, moves towards the low-pressure side with increased sealed fluid pressure, due to the fluid trying to penetrate from the high-pressure side, see Figure 6b. The variation in the length of contact at the high-pressure side, $l_h$, depicted in Figure 6c, is showing a non-monotonic increase with the sealed fluid pressure. As sidewall friction coefficient $\mu$ increases, the maximum $l_h$ occurs for a lower sealed fluid pressure. This is indicated by the dashed line in the figure. Since the gap at the low-pressure side surface is free from hydrostatic pressure, the length of contact at the low-pressure side, $l_l$, keeps increasing with the sealed fluid pressure for all five values of the sidewall friction. In the case of zero friction, the length of the low-pressure side of the contact reaches its global maximum at $p_f \approx 1.05 p_s$ where it saturates and remain constant with further increased sealed fluid pressure, as shown in Figure 6d. With increasing sidewall friction, the length of contact at the low-pressure side shows a monotonic increase as the sealed pressure increases up to $p_f = 1.2 p_s$.

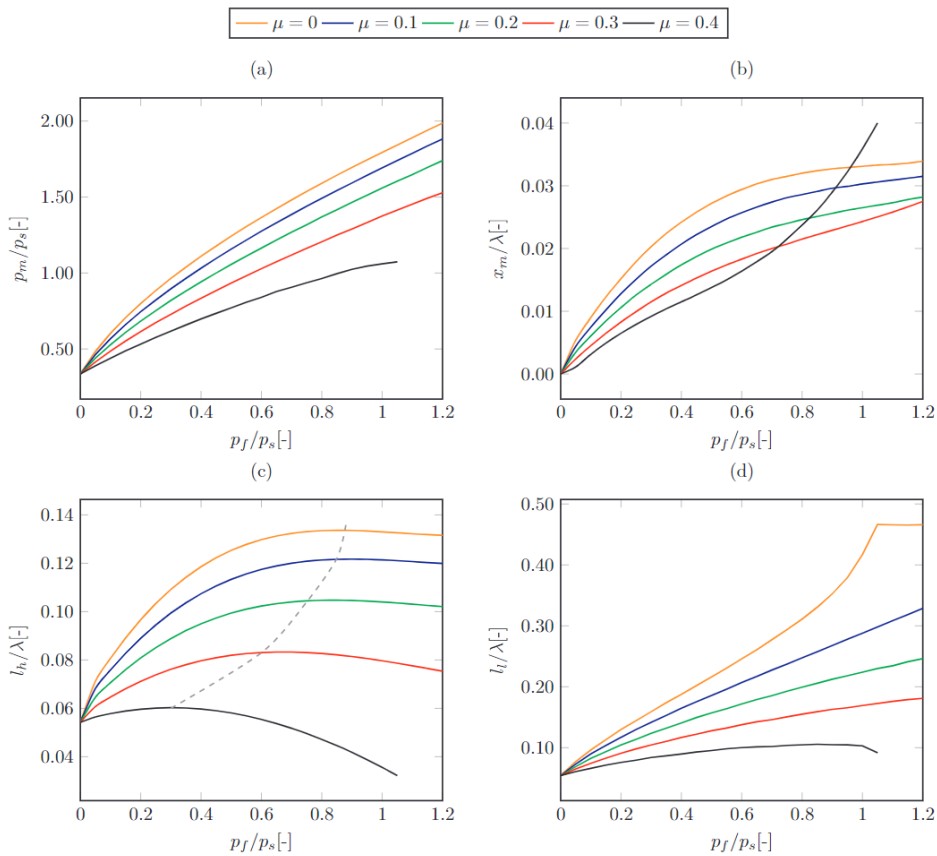

**Figure 6.** Four of the five normalized performance parameters and their variation with the sealed fluid pressure $p_f$ and the sidewall friction coefficient $\mu$. (**a**) The maximum contact pressure, (**b**) the maximum contact pressure location, (**c**) length of the contact at the high-pressure side. The dashed line indicating the position of the maximum $l_h$ for different friction coefficients, (**d**) length of the contact at the low-pressure side.

The load balance along the vertical direction for the current system is:

$$P_t = P_c + \mu\lambda p_f + p_f\left(x_m + \frac{\lambda}{2} - l_h\right), \tag{25}$$

where $P_c$ is the total contact load on $\Omega_c$. The second term in Equation (25) represents the sidewall friction and the third term is the contribution from the sealed fluid between contacting surfaces. By substituting Equation (24) into Equation (25), the dimensionless contact load, when normalized with $\lambda p_s$, can be written as:

$$\frac{P_c}{\lambda p_s} = \frac{P_0}{\lambda p_s} + K_c\frac{p_f}{p_s}, \tag{26}$$

with

$$K_c(p_f, \mu) = 0.5 - \mu - \frac{x_m}{\lambda} + \frac{l_h}{\lambda}. \tag{27}$$

For the low friction coefficient $\mu < 0.5$, the contribution of $x_m$ and $l_h$ to the slope function $K_c(p_f, \mu)$ is small compared with the term $(0.5 - \mu)$. This can be deduced from Figure 6b,c, which depict $x_m$ and $l_h$ respectively. In this case, the total contact load, $P_c$, increases linearly with the sealed fluid pressure $p_f$, as seen in Figure 7. As $\mu$ approaches 0.5, $K_c$ becomes sensitive to variations in $x_m$ and $l_h$, and the total contact load $P_c$ is no longer following the linear trend. As a function of the sealed fluid pressure, the total contact load $P_c$ for $\mu = 0.4$ flattens out when $p_f > 0.6p_s$, and $x_m$ moves towards to the low-pressure side to generate sufficient counter moment. This results in an increasing rate change of $x_m$ wtih $p_f$, as shown in Figure 6b.

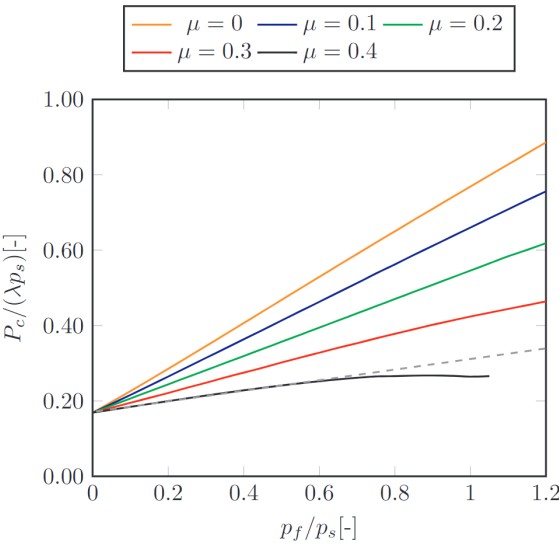

**Figure 7.** The total contact load as a function of sealed fluid pressure.

The movement of the fluid front $x_f$, starts from its initial position $x = x_0$ when $p_f = 0$, is shown in Figure 8 for the case $x_0/\lambda = -0.054$. A negative value of $x_f - x_0$ means that it has moved to the left of its initial location $x_0$. For all the cases in the current study, the location of the fluid front $x_f$ first moves towards the high-pressure side, with an increasing sealed fluid pressure $p_f$. For sidewall friction coefficients $\mu = 0.4$ and $\mu = 0.3$, there is a change in direction of movement, but for lower values of $\mu$ the change of direction is no longer observed, within the range of $p_f$ in the current study. There is a trend, however, indicating that a change of direction might occur for $p_f$ outside this range.

The overshoot pressure, $p_{over}$, initially increases towards a maximum and then it decreases with further increase in the sealed fluid pressure, $p_f$, as shown in Figure 9. The maximum contact pressure reduces with increased sidewall friction, and this is

the reason for the slower decrease in $p_{over}$. However, when sidewall friction is large, a significant decrease in $p_{over}$ can be observed, and for $\mu = 0.4$ it becomes 0 at $p_f/p_s \approx 1.05$. The simulation is terminated before $p_{over}$ reaches to zero, as the global load balance condition is violated in this case.

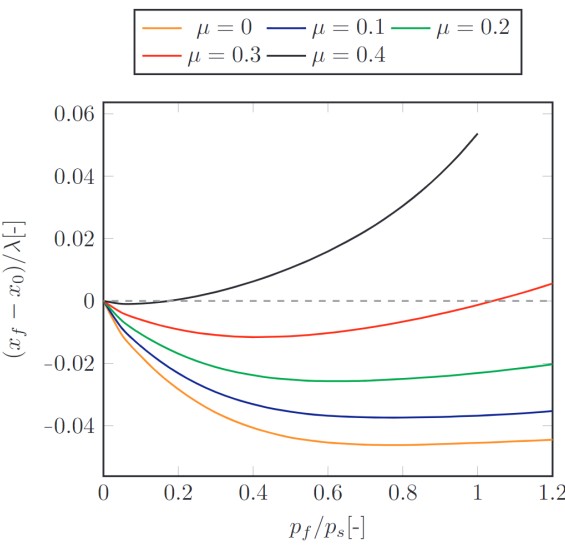

**Figure 8.** Movement of the fluid front from its initial location, as a function of the sealed fluid pressure.

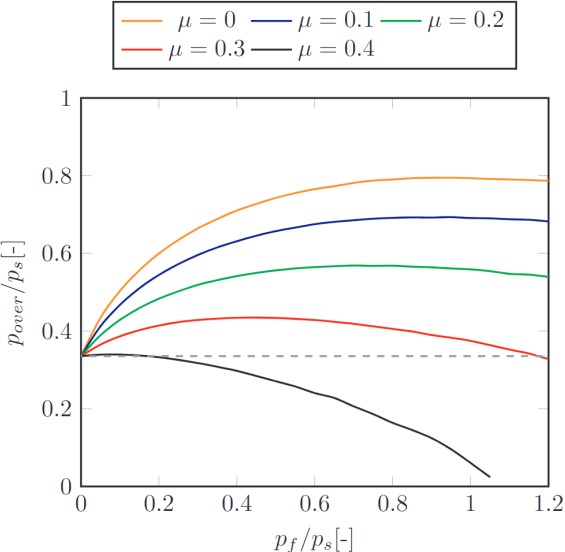

**Figure 9.** The overshoot pressure as a function of the sealed fluid pressure and the sidewall friction coefficient $\mu$.

Since the overshoot pressure is critical to determine the sealing performance and it is closely related to the total line load density $P_t = P_0 + \lambda p_f$, it is of particular interest to study the variation of overshoot pressure, $p_{over}$, with the pre-tension line load density $P_0$. It can also be studied from the perspective of the first solid-solid contact point on the left side under dry contact condition $x_0$, for intuitive understanding. The relationship of the overshoot pressure with pre-tension, for the side-wall friction coefficient $\mu = 0.4$, is depicted in Figure 10. It can be observed that the limiting case $x_0 = 0$ also shows a positive overshoot pressure. This suggests that by keeping the spacing between the seal ring top surface and the housing, during the fluid pressurization process, the seal's functionality is not compromised when there is no pre-tension apparent. However, in reality, there is surface roughness also on the vertical sidewall, and fluid can either be trapped or leak

through this interface. In turn, this would lead to a decreased sidewall friction coefficient, resulting in a higher load on the bottom sealing surface. With the increased load on the bottom surface, the seal would leak less there, and the leakage, if any, would be fluid percolating through the roughness, of either the bottom- or the side-wall interface or both. In this situation, the leakage may be estimated by means of the two-scale model presented in Reference [21].

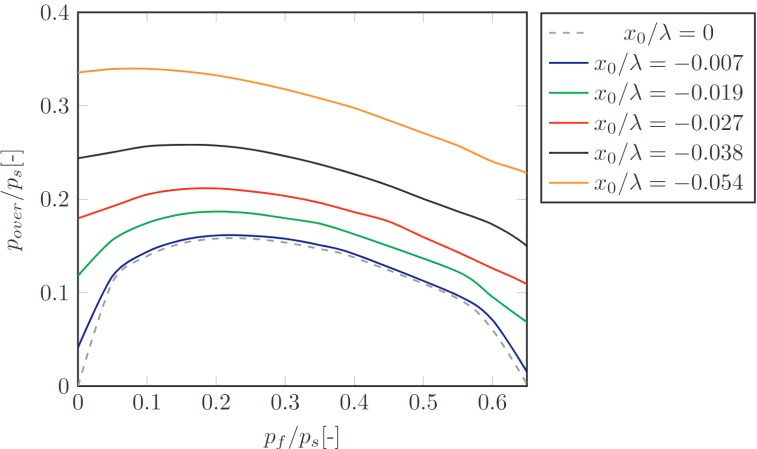

**Figure 10.** The overshoot pressure and its variation with the sealed fluid pressure and the pre-tension, for the side-wall friction coefficient $\mu = 0.4$.

The profile may have various other shapes than the cosinusoidal seal ring bottom profile given in Equation (1), which the results presented until now have been based on. It is also realized that the shape of the profile will have a rather large impact on the seal performance. To gain a better understanding and to widen the scope of the present study, a parabolic sealing surface with the same peak to valley (PV) value, that is, $(1/50)\lambda$, as the cosinusoidal profile studied before is, therefore, included in the analysis. The overshoot pressure's relationship with the sealed fluid pressure under the condition $x_0 = 0, \mu = 0.4$ is shown in Figure 11. For a total vertical line load density of $P_t = (1 - \mu)p_f\lambda$, the half contact length $l_c$ for the cosinusoidal profile [7] is

$$\frac{\pi l_c}{\lambda} = \sin\left(\sqrt{\frac{P_t}{\lambda p_s}}\right) = \sqrt{(1 - \mu)\frac{p_f}{p_s}}, \tag{28}$$

and the maximum contact pressure for the cosinusoidal profile $p_{mw}$ [7] is

$$\frac{p_{mw}}{p_s} = \frac{2P_t}{p_s\lambda}\frac{1}{\sin\frac{\pi l_c}{\lambda}} = 2\sqrt{(1 - \mu)\frac{p_f}{p_s}}. \tag{29}$$

For the parabolic profile given by Equation (2), the (maximum) Hertzian contact pressure $p_{mh}$ [22] is

$$\frac{p_{mh}}{p_s} = \frac{1}{p_s}\left(\frac{16\Delta P_t E}{\pi\lambda^2(1 - \nu^2)}\right)^{\frac{1}{2}} = \frac{4}{\pi}\sqrt{(1 - \mu)\frac{p_f}{p_s}}, \tag{30}$$

and we notice that the maximum contact pressure, for both profiles, follows a square root increase with the sealed fluid pressure. Moreover, the relationship $p_{mw}/p_{mh} = \pi/2$, is valid for the case when there is no fluid present, that is, the dry contact case. Initially, as the sealed fluid pressure increases, the overshoot pressure for both profiles increases, and the ratio of $p_{over}$ between the two profiles is approximately $\pi/2$ up to $p_f = 0.05p_s$, due to the small contribution of the sealed fluid pressure. At the value of $p_f = 0.05$, the overshoot pressure for the parabolic profile is almost at its highest value, and thereafter it decreases. A significantly smaller sealed fluid pressure is required for the overshoot pressure to vanish

for the parabolic profile, than for the cosinusoidal profile. This suggests that a seal with a parabolic profile is prone to lose its functionality at an earlier stage than if the profile would be cosinusoidal, with the same PV value.

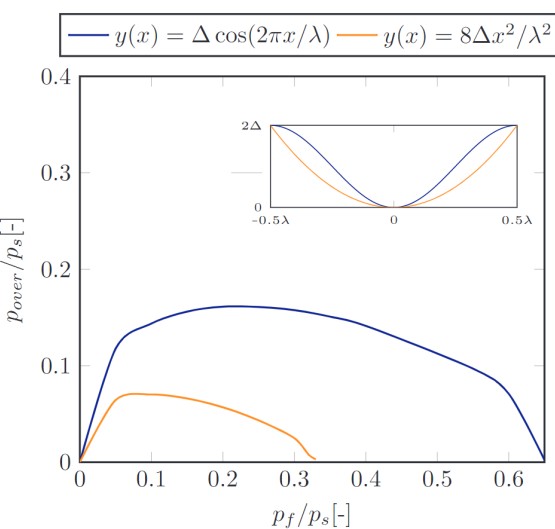

**Figure 11.** Dimensionless overshoot pressure as a function of non-dimensional fluid pressure, for a cosinusoidal (**blue line**) and a parabolic (**orange line**) bottom surface profile.

## 4. Concluding Remarks

We have presented a finite element based model that resembles the static seal assembly and we have studied its functionality under various conditions. The seal assembly is modeled as a contact mechanics problem including the hydrostatic load at the contacting interface, caused by the sealed fluid pressure. The model is verified against available results for cosinusoidal wavy surfaces with and without fluid entrapment inside the undulating gap between the contacting surfaces.

Five sealing performance parameters to describe the contact between sealing surfaces and the shape of the corresponding pressure distribution were defined. These are (i) the value of the maximum normal stress, (ii) the location of the maximum normal stress, (iii) the length of contact at the high-pressure side, (iv) the length of contact at the low-pressure side, and (v) the overshoot pressure, that is, the difference between maximum contact pressure and the sealed fluid pressure.

The sealing performance of seal rings with both cosinusoidal and parabolic profiles was studied, and the results show for example, that there is a $\pi/2$-relationship between the overshoot pressures, for the two profiles, in the lower range of sealed fluid pressures. The results also suggest that the functionality of the seal is compromised at a significantly much lower sealed fluid pressure for the parabolic profile, than for the cosinusoidal one.

The movement of the fluid front has also been studied, and the results suggest that the functionality of seal will only be compromised if the sidewall friction is large enough. For lower values of the sidewall friction coefficient, the sealed fluid pressure must be unrealistically much higher.

**Author Contributions:** D.H.: Conceptualization, Methodology, Data Curation, Writing—Original Draft. X.Y.: Resources, Writing—Review & Editing. R.L.: Project administration, Writing—Review & Editing. A.A.: Supervision, Writing—Review & Editing. All authors have read and agreed to the published version of the manuscript

**Funding:** the Swedish Research Council (Vetenskapsrådet), via the project: Multiscale topological optimisation for lower friction, less wear and leakage, with registration number 2017-04390.

**Conflicts of Interest:** The authors declare no conflicts of interest.

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
