# Peer review of "Numerical Simulation of Static Seal Contact Mechanics Including Hydrostatic Load at the Contacting Interface"

_lubricants, doi:10.3390/lubricants9010001_

Round 1

Reviewer 1 Report

  1. This research used COMSOL 5.4 to study the effects of preload, sealed fluid pressure and static friction coefficient on the characteristics of a static seal. It claims, as on the title, including fluid-structure interaction. Nevertheless, there is no numerical work on solving the Navier-Stokes equations. Also, the title highlights the “finite-element model” implying that the contribution is on the model development. Yet, no validation of this model, and the whole work is to find out how static seals characteristics under various condition. The authors need to choose a better title for this article.
  2. Many important contents about the figures were stated in the figure captions. Yet, readers might want to find out what the figures are by reading the main body paragraph. It’s better to briefly state them in while referring to a figure.
  3. Some important messages were missing. For example, what is the Westergarrd solution? What is lambda prime in Eq. (22)? Equation 22 is for what condition, e.g., what’s the deformation of the seal face or what’ the sized of the deformed seal face etc. Where did Eq. (23)come from?
  4. The Eq. (15) is especially difficult to understand. Why (sigma dot n) is less than –p_f?
  5. The contact pressure p_c is important to the tribologists. The article should discuss about it.
  6. The equations (24)—(26) are very confusing. The reviewer has to read them several times to understand the equations and the purpose of stating them that way. The authors might need to write more paragraphs to help the readers.
  7. From lines 171 to 173, the article claims “… when p_f > 0.6…” Why? Where does this “0.6” come from? Besides, p_f is a pressure. There should be a unit.
  8. In the caption of Fig. 5, it says “… indicating the position of the maximum for …” Position should be location. And maximum of “what”?
  9. Deformation is an important physical quantity that we usually like to see. The reviewer recommends to discuss it.
  10. Discussions about the derivation for Eqs. (27) to (29) are required. At least, a citation is a must if they come from the literature.
  11. Many symbols should be explained clearly. For example, l_c0? What it is? “initial” half-contact length associated with “what”?
  12. Word choice: For example, in abstract, lines 2 to 4, the “controlled” should be “balanced”, just to name a few.

Reviewer 2 Report

Static sealing is wildly used in machine element, its contact condition is important to the sealing performance. Authors developed a finite element model of static seal element and different sealing performance parameters were used to describe the contact pressure. It cannot be acceptable in current paper. Some comments are given by:

  1. In Introduction part, many researchers have studied the sealing performance in static seal. Here, the research status of related sealing performance at home and abroad is insufficient.
  2. In Problem set-up part, seals is deformable, such as rubber ring, which is shown in Fig 1. How to consider the deformation? Why the influence of microscopic morphology of housing sidewall on the contact deformation is not considered?
  3. In Results part, In Figure 5, what are the considerations for choosing the range of friction coefficient?It is depending on the contact condition of the surfaces.
  4. More details of finite element modeling for sealing should be present.

Reviewer 3 Report

It is a good work as for a purely numerical parametric study of leakage considering idealised fluid-structure interaction. At least some amount of experimental validation would make this research much more convincing.

Round 2

Reviewer 1 Report

Please see the attached word file.

Reviewer 2 Report

Static sealing is wildly used in machine element, its contact condition is important to the sealing performance. Authors have revised the manuscript according to the commands. However, It cannot be acceptable in current paper.

  1. In line 84, “A linear elastic material is assigned to the seal”Here, the elastic modulus of the seal should be given, and its reference to the material parameters.

  1. In line 105, “This resulted in meshes with 8367, 16467 and 32253 regular- and 682, 1210 and 2236 edge elements, respectively.” Here, the meshing model by COMSOL can be shown in the paper, including the boundary conditions. It is better to give a schematic diagram of the model.

  1. Seen in Fig 3 , why is the contact deformation asymmetric in dry contact?

Round 3

Reviewer 1 Report

line 145: "The solid lines in Figure 4 represent..." should be "The solid lines in Figure 4(b) represent..."